# Stomatal Ratio Showing No Response to Light Intensity in *Oryza*

**DOI:** 10.3390/plants12010066

**Published:** 2022-12-23

**Authors:** Tiange Wang, Linna Zheng, Dongliang Xiong, Fei Wang, Jianguo Man, Nanyan Deng, Kehui Cui, Jianliang Huang, Shaobing Peng, Xiaoxia Ling

**Affiliations:** National Key Laboratory of Crop Genetic Improvement, Hubei Hongshan Laboratory, MOA Key Laboratory of Crop Ecophysiology and Farming System in the Middle Reaches of the Yangtze River, College of Plant Science and Technology, Huazhong Agricultural University, Wuhan 430070, China

**Keywords:** growth environments, stomatal density, stomatal ratio, photosynthesis, rice

## Abstract

Stomata control carbon and water exchange between the leaves and the ambient. However, the plasticity responses of stomatal traits to growth conditions are still unclear, especially for monocot leaves. The current study investigated the leaf anatomical traits, stomatal morphological traits on both adaxial and abaxial leaf surfaces, and photosynthetic traits of *Oryza* leaves developed in two different growth conditions. Substantial variation exists across the *Oryza* species in leaf anatomy, stomatal traits, photosynthetic rate, and stomatal conductance. The abaxial stomatal density was higher than the adaxial stomatal density in all the species, and the stomatal ratios ranged from 0.35 to 0.46 across species in two growth environments. However, no difference in the stomatal ratio was observed between plants in the growth chamber and outdoors for a given species. Photosynthetic capacity, stomatal conductance, leaf width, major vein thickness, minor vein thickness, inter-vein distance, and stomatal pore width values for leaves grown outdoors were higher than those for plants grown in the growth chamber. Our results indicate that a broad set of leaf anatomical, stomatal, and photosynthetic traits of *Oryza* tend to shift together during plasticity to diverse growing conditions, but the previously projected sensitive trait, stomatal ratio, does not shape growth conditions.

## 1. Introduction

The microscopic stoma pores on the leaf surface control the rate of CO_2_ and vapor exchange between the internal leaf airspace and the atmosphere. The diffusion efficiency via stomatal pores was generally measured as stomatal conductance to water vapor (g_sw_). As water and CO_2_ share the same gaseous diffusion pathway via stomata, the stomatal conductance to CO_2_ (g_sc_) can be calculated from g_sw_ divided by 1.6, the ratio of the diffusivities of CO_2_ to water vapor in the air. Stomatal conductance (g_s_) is a key determinant of plant productivity and water use efficiency since it balances CO_2_ uptake and water loss. A pair of guard cells form each stoma, and stomates have many shapes, numbers per leaf area, and sizes [1,2,3].

Stomata are morphologically diverse across plants [4]. For instance, the stomatal complexes of eudicots are typically formed with kidney-shaped guard cells, but the stomatal complexes of monocot grasses are formed with dumbbell-shaped guard cells flanked by a pair of subsidiary cells. Several recent studies reported that the g_s_ of species equipped with dumbbell-shaped guard cells showed a faster response to environmental changes than those equipped with kidney-shaped stomata [5,6]. Although the stomatal shape-determining functions have been discussed frequently, only Franks and Farquhar [5], to the best of our knowledge, provided direct evidence by comparing the mechanical property of stomatal complexes of eudicots and grasses in response to light and air humidity changes. Beyond stomatal shape, stomatal patterning also differed significantly between eudicots and grasses. Stomata are scattered throughout the epidermis in eudicots but in parallel rows within epidermal cell files in grasses [4,7].

Stomatal size and density and their impacts on g_s_ have been extensively studied [8,9,10]. In steady-state environments, g_s_ is tightly correlated to both stomatal size and density, which also influence the g_s_ kinetics in response to environmental changes [11,12,13,14,15]. Stomatal size and density are dramatically impacted by growth environment factors, including light intensities, water stress, and CO_2_ concentration elevation [16,17,18,19]. For instance, the smaller but denser stomata over the evolutionary period were related to the adaption of the species to lower atmospheric CO_2_ concentrations [20]. However, these observations were primarily for eudicots, with not many for grasses, including all cereal crops, which play a crucial role in human society [21,22]. Detailed knowledge about the stomatal traits in accommodating to different growth environments, therefore, is necessary to improve cereal crop photosynthesis and grain yield.

The allocation of stomata between adaxial and abaxial surfaces of leaves significantly impacts stomatal conductance and photosynthesis [3,23]. Although most eudicot species are hypostomatous, in which stomata are only distributed on the abaxial surface, cereals are amphistomatous, of which stomata are allocated on both leaf surfaces [3,24,25]. Several recent studies suggested that amphistomatous species have advantages in CO_2_ diffusion from leaf surface to chloroplasts via improved boundary layer conductance, stomatal conductance, and mesophyll conductance simultaneously [3,24,26]. In amphistomatous, it was suggested that stomata developed independently on the two surfaces [27]. Studies have shown that stomata on two leaf sides respond differently to environmental changes in some species [28] and the stomatal density ratio of the abaxial and adaxial surface, termed the stomatal ratio, was observed to vary with growth environments in many species [27,29,30]. Again, these studies focused mostly on eudicot species, and the field has largely ignored monocots. Eudicot leaves are typically bifacial, where mesophyll cells within the upper and lower portions of the leaf are differentiated into palisade and spongy mesophyll tissue [28,31]. However, monocot leaves tend to be isobilateral, where the abaxial and adaxial tissues are symmetrical.

Exploiting the mechanisms underlying the responses of stomata allocation on leaf surfaces to environmental changes, alongside other stomatal traits, may provide us with a potential approach to improve cereals photosynthesis. In this study, we grew five *Oryza* species in two environmental conditions, an environmentally controlled growth chamber, a widely used facility in plant science research, and an outdoor one, to answer the following questions: (1) How do stomatal features, including stomatal size, stomatal density, and stomata allocation on leaf surfaces, vary among *Oryza* species? (2) Will acclimation to growth environments result in different stomatal features leading to stomatal conductance and photosynthetic capacity?

## 2. Results

### 2.1. Variation across Species in Leaf Traits

As shown in Figure 1, the active photosynthetic radiation, air temperature, and relative humidity were relatively constant in the growth chamber but showed considerable variation over the day and during the growth period in the outdoor condition. The daily active photosynthetic radiation was much lower in the growth chamber than in the outdoor condition. We observed substantial variation across the Oryza species in leaf anatomy, stomatal traits, the photosynthetic rate, and stomatal conductance (Table 1 and Figure 2). Upon averaging trait values for each species across the two growth conditions, leaf width varied 1.83-fold, major vein thickness varied 2.07-fold, minor vein thickness varied 1.51-fold, and major vein density varied 1.70-fold (Appendix A). Oryza species varied 2.01-fold, 2.12-fold, and 1.24-fold in abaxial stomatal density, adaxial stomatal density, and the stomatal ratio, respectively (Figure 3 and Appendix A). Species varied 1.35- to 1.77-fold in stomatal morphological traits and 1.70- to 1.77-fold in photosynthetic rate and stomatal conductance. The trait variations were larger across species grown in outdoor conditions than in the growth chamber except for minor vein thickness, the stomatal ratio, and gas exchange (Appendix A).

### 2.2. Plasticity across Growth Conditions in Leaf Traits

On average, photosynthetic capacity, stomatal conductance, stomatal density on both adaxial and abaxial surfaces, leaf width, major vein thickness, minor vein thickness, inter-vein distance, and stomatal pore width values for leaves developed by plants grown outdoors were higher than those for plants grown in the growth chamber (Figure 4). No differences were observed in the stomatal ratio, major vein density, stomatal complex width, and guard cell width for plants grown in the two conditions. In contrast, minor vein density and the adaxial stomatal pore length for plants grown outdoors were lower than those in the growth chamber. Interestingly, the plasticity responses of leaf traits to growth conditions were species dependent (Figure 5). Across species, only the adaxial stomatal density and major vein thickness values for leaves developed by plants grown outdoors were consistently higher than those for plants grown in the growth chamber. Unexpectedly, neither stomatal conductance nor the photosynthetic rate was correlated with stomatal density across species and growth environments (Appendix A). For *O. australiensis*, six of the twenty-three leaf traits showed a plasticity response to growth environments, and sixteen of the twenty-three leaf traits differed significantly between the two growth conditions for *O. sativa*, the cultivated species. We also found that leaf trait plasticity response to growth environments differed in magnitude and even direction from each Oryza species (Figure 5). Overall, the stomatal density, photosynthetic rate, and stomatal conductance were correlated between plants grown in the growth chamber and outdoor conditions (Figure 6).

### 2.3. Upper versus Lower Leaf Surface

The leaves of *Oryza* species are amphistomatous, which stomates allocate on both adaxial and abaxial leaf surfaces. The abaxial stomatal density was higher than the adaxial stomatal density in all the species, and the stomatal ratios ranged from 0.35 (*O. minuta*) to 0.46 (*O. australiensis*) across species in two growth environments. No difference in the stomatal ratio was observed for a given species between plants in the growth chamber and outdoors (Appendix A). The stomatal densities of adaxial and abaxial leaf surfaces showed a similar response to growth environment changes and were well correlated across species and growth environments (Figure 7a). The stomatal size represented by the stomatal length on two leaf surfaces was similar in each species, unlike stomatal density, and the abaxial and adaxial stomatal sizes were well correlated across species and growth environments (Figure 7b). Light response curves were measured in plants growing in the growth chamber and outdoors in two ways, by illuminating the upper or lower surfaces (Appendix A). Photosynthetic rates in high light conditions (i.e., PAR > 500 mmol m^−2^ s^−1^) measured by illuminating the upper surface were higher than by illuminating the lower surface in all species and growth conditions except for *O. australiensis* in the growth chamber.

## 3. Discussion

### 3.1. Stomatal Density Variation among Species

The stomatal conductance and thus photosynthesis has been widely suggested to be determined by stomatal morphological features, including stomatal density, size, and pore aperture. Among those traits, the impacts of stomatal density on rice stomatal conductance and photosynthesis have been extensively studied. However, studies on rice have found positive, negative, or even no correlations between stomatal conductance and stomatal density [21,32,33]. The current study found that the variations of photosynthetic rate and stomatal conductance across *Oryza* species and growth conditions were independent of stomatal density (Appendix A). There are several reasons for the lack of a relationship between stomatal density and gas exchange parameters. Firstly, the stomatal conductance was relatively high in all the estimated species in the two growth conditions, and the stomatal conductance might be large enough to support photosynthesis. Recent studies have shown that the photosynthesis of C3 crops is mainly limited by photosynthetic biochemistry rather than CO_2_ diffusion conductance [6,34]. Secondly, stomata density on the leaf surface may constrict stomatal conductance by increasing overlaps between diffusion shells of neighboring stomatal pores, as confirmed by Lehmann and Ort [35]. Thirdly, patchy stomatal closure might occur when the leaf evaporation is high, and the apparent leaf conduction may decline [36]. Obviously, further studies are required to fully understand why the increase in stomatal pores does not result in a higher stomatal conductance in rice.

### 3.2. Responses of Stomatal Density, Size, and Ratio to Growth Conditions

Salisbury [37] first reported in 1928 that the stomatal density of the same species was higher in plants grown in sunny conditions than in shaded conditions. To date, it is clear that both intrinsic and external environmental factors regulate stomata development [14]. Among the environmental factors, CO_2_ concentration and light intensity were often considered the dominant controls of stomatal development. The current study found that the stomatal density was higher in plants grown in outdoor conditions than in the growth chamber. Although the average light intensity was much lower in the growth chamber than in the outdoor conditions, temperature, humidity, light quality, and environmental factor fluctuations were also different between the two growth conditions. Therefore, the main environmental factors controlling the stomatal density of *Oryza* species cannot be identified in the current study.

Interestingly, the domesticated species, *O. sativa*, showed the most prominent response ratio in stomatal density to environmental changes (Figure 5), indicating that the plasticity response of stomatal density might be one of the domestication-related traits in rice. It has been suggested that the stomata on adaxial and abaxial leaf surfaces develop independently, and the stomatal ratio is regulated by light intensity and water available to eudicot species [23,38,39,40]. Although the water was adequate in both conditions, the light environments in the growth chamber and outdoor conditions are different in intensity, fluctuation, and quality. Unexpectedly, the stomatal ratio was not shaped by growth environment changes in all the investigated *Oryza* species (Appendix A and Figure 5). The variation of the stomatal ratio across species was in a narrow range, and, overall, approximately 60% of the stomata were allocated on the abaxial surface. The constant stomatal ratio across *Oryza* species and growth environments might be related to the pattern of the stomatal distribution on the leaf surface. In fact, the stomata on *Oryza* leaves are distributed in parallel rows along the veins on both adaxial and abaxial surfaces rather than being scattered throughout the leaf surfaces as in eudicots. There is always one row in adaxial surfaces and two rows in abaxial surfaces, and the distribution pattern contributed to the robust stomatal ratio. We divided these leaf traits into demand-related traits and supply-related traits according to their function in photosynthetic gas exchange. The demand traits such as SD_ab_ and SD_ad_ were found to be higher in outdoor conditions than in the growth chamber (Figure 4), which means that the gas exchange demand increased under high light. Considering neither major nor minor vein density contributed to the balance between supply and demand and the inter-vein distance increased under high light (Figure 4), we concluded that it was the increased vein thickness that worked. The decrease in the minor vein density and the increase in the leaf width and inter-vein distance together indicated that the mesophyll tissue increased due to the wider distance between the two veins and the higher transportation capacity of the conduit (Table 1 and Figure 4). However, the non-increased vein density in those five species indicates the existence of vascular redundancy, which represents an overweight leaf vein [41].

In the current study, the stomatal size was represented using the stomatal length since it changes only slightly when stomates open and close [42]. In contrast to stomatal density, the stomatal size on the adaxial surface was lower in plants growing outdoors than in the growth chamber. It should be noted that the difference in the adaxial stomatal size between plants growing in the growth chamber and outdoors is much smaller than the difference in stomatal density. The reduced stomatal size in outdoor conditions is expected as a small stomatal size has been proposed to facilitate a faster aperture response in fluctuating light environments [11,13]. However, we found that the abaxial stomatal size showed no response to the growth environments of the plants. The different responses of adaxial and abaxial stomatal sizes to growth conditions may relate to the light environment on the two surfaces. Unlike adaxial surfaces typically exposed to direct light, whether the plants were growing in the growth chamber or outdoors, the leaf blade itself shades the abaxial leaf surfaces and stays in relatively low-light conditions. Indeed, it has been reported that the abaxial surface only received approximately 10% of the total light incident [43,44].

### 3.3. Growth Chamber versus Outdoor

Growth chambers are widely used to deliver consistent, repeatable growing conditions to help drive the discovery process. However, as shown here, the environmental conditions in the growth chambers differed significantly from the natural conditions, and it is not surprising that the absolute values of leaf anatomical and physiological traits differed. Therefore, an interesting question arises: Can the plant trait variations measured in the growth chamber reflect their variations outdoors? A few studies have explored this question, and the results are mixed [45]. Here, we found that the stomatal trait and physiological traits estimated on plants growing in the growth chamber were generally well correlated with the estimations on rice plants growing outdoors (Figure 6). We should note that the outdoor plants were potentially impacted by many random fluctuations in the local climate, which may influence the comparison [46]. Notably, there are significant interactions between growth conditions and species. Among the 23 estimated traits, only the major vein thickness and adaxial stomatal density were changed in the same direction when comparisons were made between the rice plants growing in the growth chamber and outdoors. Clearly, the translation of the observations from the growth chamber to outdoors is tough in most cases. As a large number of plant science studies were and will be continually conducted in controlled conditions, more efforts should be paid to bridge the gap between growth chambers and outdoor investigations.

## 4. Materials and Methods

### 4.1. Plant Material and Growth

After the previous measurement of 47 *Oryza* species, five *Oryza* species that have a significant difference in stomatal traits, namely, *O. australiensis*, *O. glumaepatala*, *O. minuta*, *O. punctata*, and *O. sativa* (c.v. Huanghuazhan), were grown in 13.0 l pots filled with 10 kg of dry paddy soil. We prepared sixteen pots with a density of one plant per pot for each species, i.e., a total of 16 plants per species were used. Half of the pots were placed outdoors on the campus of Huazhong Agricultural University (30.46° N, 114.36° E), and the other half of the pots were placed in the growth chamber (Conviron GR144, Controlled Environments Lit., Manitoba, Canada). All plots were placed at a distance of circa 40 cm. The photosynthetic active radiation (PAR), air temperature, and relative humidity in the growth chamber and outdoors over the duration of the experiment are shown in Figure 1. Among these environmental factors, the difference in PAR between growth environments was the biggest, and other factors can be ignored. Before planting, 5.0 g of compound fertilizer (N: P_2_O_5_: K_2_O = 1:1:1) was mixed into the soil for each pot. The plants were watered daily, and the weeds and pests were well controlled. Pots were rearranged weekly to avoid edge effects, and measurements were conducted at the active tillering stage, i.e., fifty days after planting.

### 4.2. Light Response Curves

All the measured leaves were grown under setting conditions until the tillering stage, and then the light response curve of photosynzfthesis was measured using an LI-6800 Portable Photosynthesis System (LI-COR, Lincoln, NE, USA). To minimize the influence of fluctuating environmental factors on photosynthetic gas exchange, the gas exchange measurement was carried out indoors with well-controlled environmental conditions. The leaf chamber conditions were the reference CO_2_ concentration of 400 µmol mol^−1^, relative humidity of 65%, and leaf temperature of 28 °C. The light levels were 1800, 1500, 1200, 1000, 800, 600, 300, 200, 100, 50, and 0 µmol m^−2^ s^−1^ PAR with 3–5 min stabilization between each light level. For each individual plant, two light response curves were measured with one leaf irradiated on the adaxial surface and another leaf irradiated on the abaxial surface. The measurement was conducted on new and fully expanded leaves, and for each treatment, at least five individuals were measured.

### 4.3. Stomatal Density and Size Measurements

Small leaf discs were removed from the middle of each leaf section as described in Xiong et al. [47]. For each species, five leaves from different plants were sampled. Leaf discs were stored in a pentanediol fixing solution at 4 °C until analysis. To measure the stomatal density, at least 30 representative areas of the abaxial or adaxial epidermal surfaces for each sample were captured at 400× magnification using a scanning electron microscope (JSM-6390LV, Tokyo, Japan). In the current study, the stomatal morphological traits including stoma pore length, pore width at the center of the stoma, and the guard cell width at the center of the stoma on the abaxial and adaxial lamina surface were also measured (Appendix A). To measure the morphological traits, stomata images were taken at 3000× magnification, and forty stomata were measured for each species. Image analysis was conducted in Image J software (National Institute of Health, Bethesda, MD, USA).

### 4.4. Leaf Anatomical Traits

The leaf pieces were infiltrated with Formaldehyde Alcohol Acetic Acid (FAA; Formalin, Acetic acid, 70% Alcohol) solution for 72 h at 4 °C. Samples were dehydrated in an ethanol series and embedded in paraffin before cutting using a fully automated rotary microtome (Leica RM2265, Leica Microsystems, Milton Keynes, UK). The leaf sections were stained with 1% (*w*/*v*) toluidine blue in 1% (*w*/*v*) Na2B4O7, and they were examined at 40× and 60× magnification with an Olympus IX71 light microscope (Olympus Optical, Tokyo, Japan). Inter-vein distance, major vein density, minor vein density, and vein thickness were measured using cross-section images [48].

### 4.5. Statistical Analysis

The light response curve was fitted to the non-rectangular hyperbola-based model:$$A=\frac{\phi \times PAR+A_{max}-\sqrt{{\left(\emptyset \times \mathrm{PAR}+A_{max}\right)}^{2}-4\theta \times \emptyset \times \mathrm{PAR}\times A_{max}}}{2\theta }-R_{n}$$
where *A* is the net photosynthetic rate, *ϕ* is the quantum yield at PAR = 0 μmol (photon) m^−2^ s^−1^, *A_max_* is the maximum gross photosynthetic rate, *θ* is the convexity factor, and *R*n is dark respiration. The model was fitted to the data using the Orthogonal Nonlinear Least-Squares Regression (onls) function. Parametric ANOVA was performed using packages of agricolae. Other analyses and plots were conducted using the tidyverse package. All analyses were performed on the R 3.6.3 platform [49].

## Figures and Tables

**Figure 1 plants-12-00066-f001:**
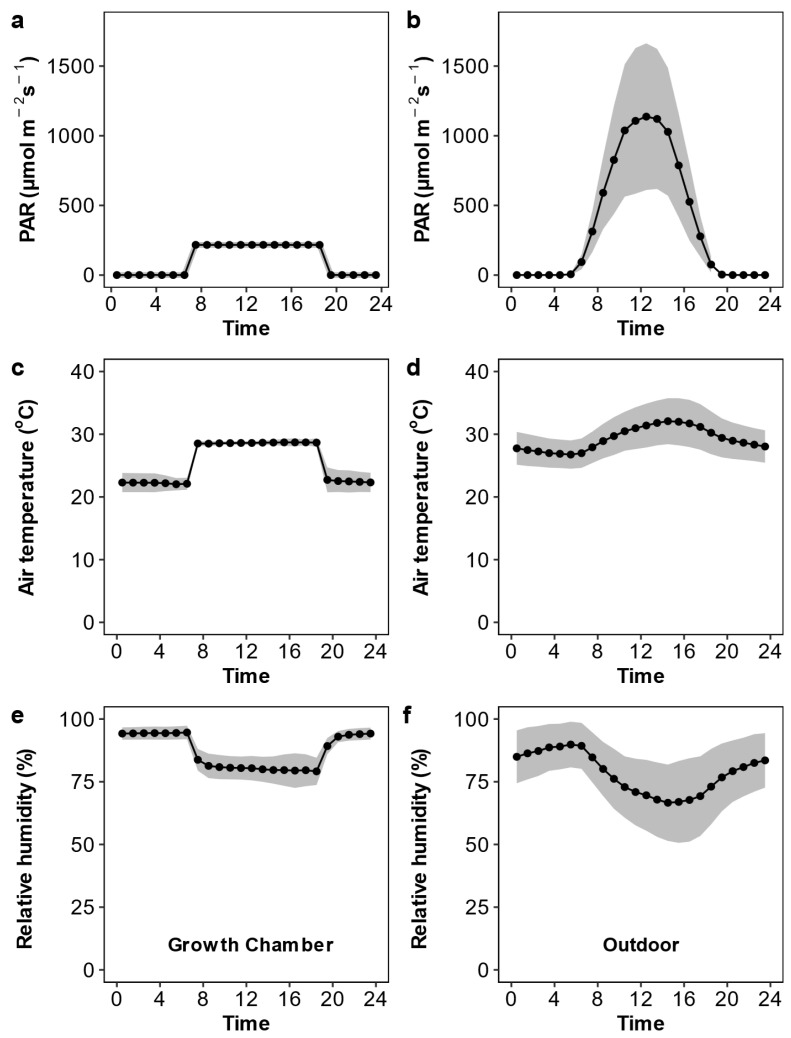
Photosynthetic active radiation (PAR; **a**,**b**), air temperature (**c**,**d**), and relative humidity (**e**,**f**) inside the growth chamber (**a**,**c**,**e**) and outdoors (**b**,**d**,**f**). The black points and lines are the mean values over the growth duration, and grey shaded areas are the 95% confidence interval.

**Figure 2 plants-12-00066-f002:**
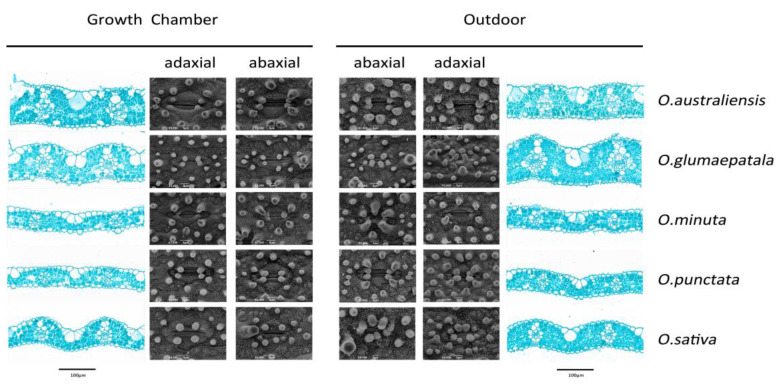
Plasticity of leaf anatomy and stomatal morphology in response to growth conditions for five *Oryza* species.

**Figure 3 plants-12-00066-f003:**
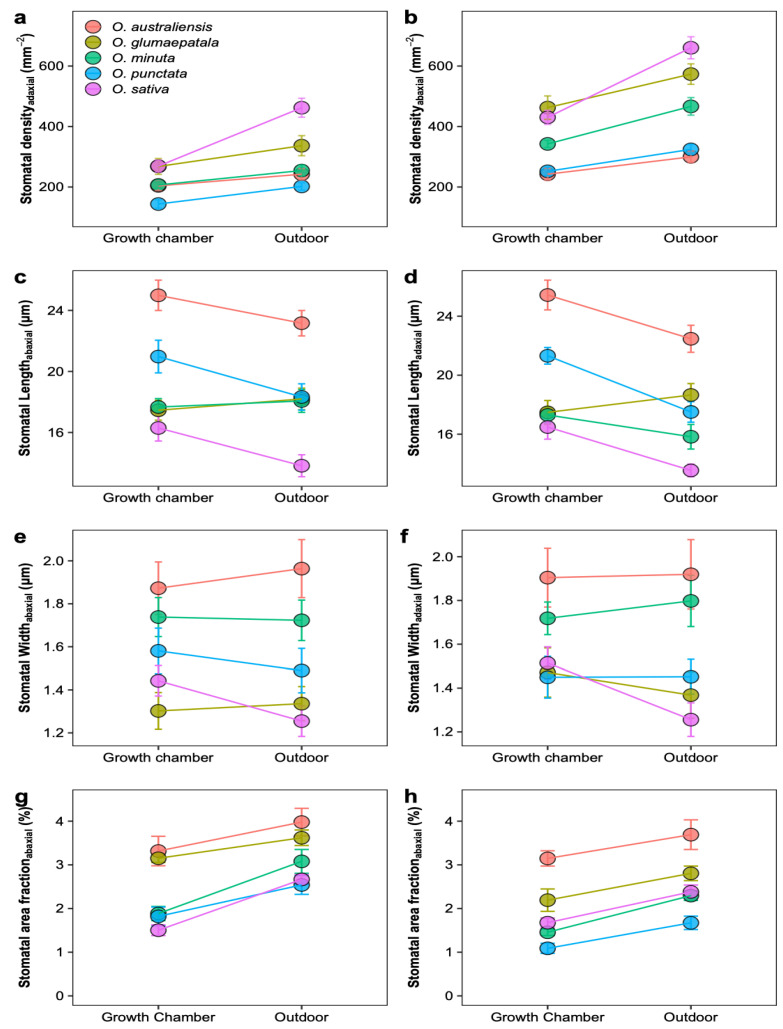
Plastic response of stomatal traits to growth environments for five *Oryza* species. Mean ± SE values for stomatal density (**a**,**b**), stomatal pore length (**c**,**d**), guard cell width (**e**,**f**), and stomatal area fraction (**g**,**h**). All the traits were estimated on adaxial (**a**,**c**,**e**,**g**) and abaxial (**b**,**d**,**f**,**h**) surfaces separately. All traits showed significant variation across species (*p* < 0.01, ANOVA); *n* = 5 individual plants.

**Figure 4 plants-12-00066-f004:**
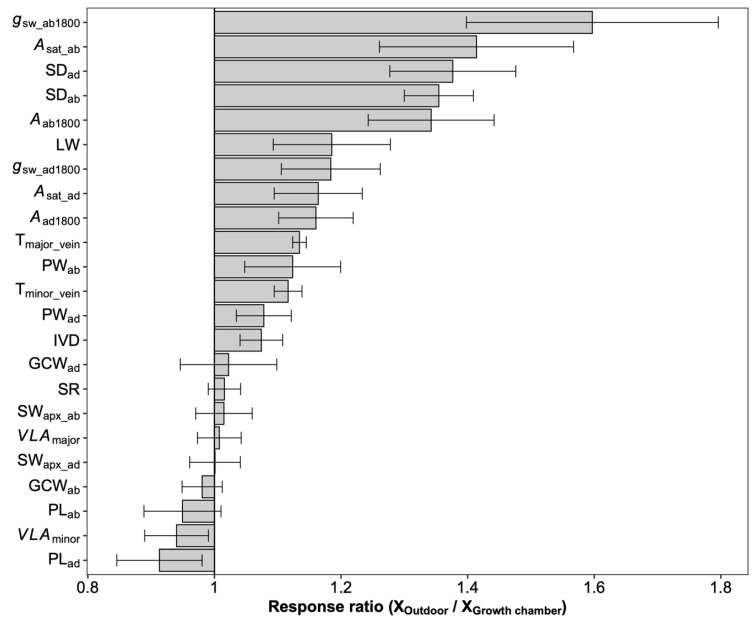
The overall response ratio of leaf traits to growth environments. Bars denote the response ratios, and the error bars are the 95% confidence intervals (calculations see M&M section). A response ratio smaller than, equal to, and larger than 1 corresponds to a lower, equal, and higher value under outdoor conditions than under growth chamber conditions, respectively. The response ratios statistically differed from 1 if their confidence intervals did not cover 1. See Table 1 for definitions of the variables.

**Figure 5 plants-12-00066-f005:**
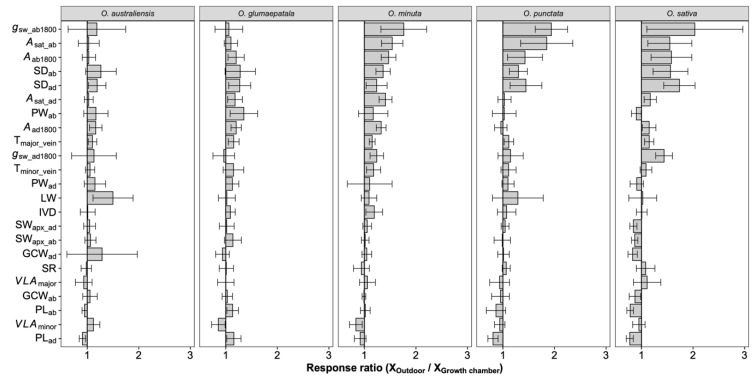
Response ratio of leaf traits to growth environments for each species. Bars denote the response ratios, and the error bars are the 95% confidence intervals (calculations see M&M section). A response ratio smaller than, equal to, and larger than 1 corresponds to a lower, equal, and higher value under outdoor conditions than under growth chamber conditions, respectively. The response ratios statistically differed from 1 if their confidence intervals did not cover 1. See Table 1 for definitions of the variables. *n* = 5 individual plants.

**Figure 6 plants-12-00066-f006:**
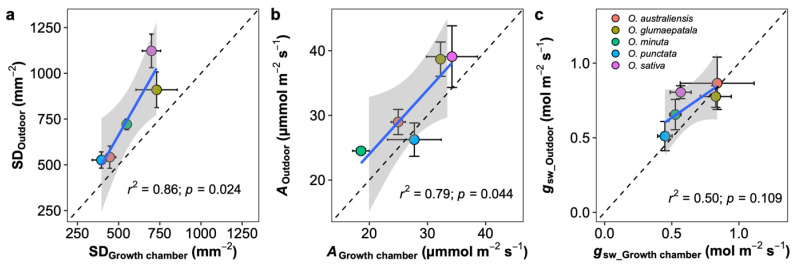
Leaf traits comparison between the outdoor and growth chamber plants. (**a**) Stomatal density per leaf area (SD), (**b**) net photosynthetic rate (A), and (**c**) stomatal conductance to vapor (gsw). Blue line represents the linear regression line; black dot line represents the 1:1 line; and the grey shaded area corresponds to the 95% confidence region of the regression line. Data are shown with mean ± SE; *n* = 5 individual plants.

**Figure 7 plants-12-00066-f007:**
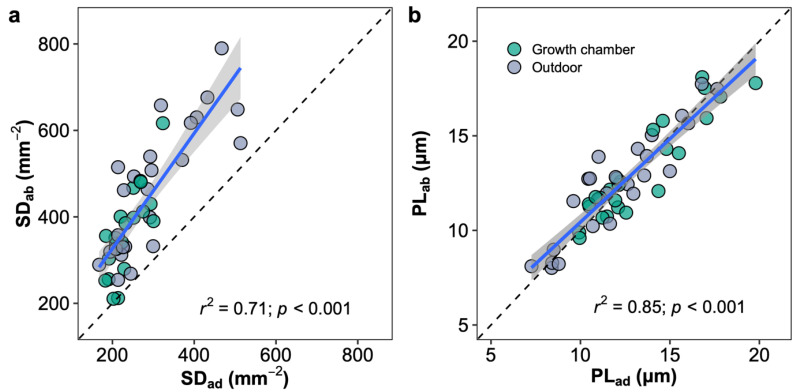
Correlations of stomatal density (SD; **a**) and pore length (PL; **b**) between adaxial (ad) and abaxial (ab) surfaces. Blue line represents the linear regression line; black dot line represents the 1:1 line; and the grey shaded area corresponds to the 95% confidence region of the regression line.

**Table 1 plants-12-00066-t001:** Leaf traits for five Oryza species, with expected plastic and adaptive responses to growth conditions (outdoors and in the growth chamber), and results of the ANOVA testing the effects of species differences, growth condition, and their interaction. ns, *p* > 0.05; *, *p* < 0.05; **, *p* < 0.01; ***, *p* < 0.001.

Trait	Symbol	Unit	Species	Growth Condition	Species × Growth Condition
Adaxial stomatal density	SD_ad_	mm^−2^	***	***	***
Abaxial stomatal density	SD_ab_	mm^−2^	***	***	**
Stomatal density	SD	mm^−2^	***	***	***
Minor vein thickness	T_minor_vein_	μm	***	***	ns
Major vein thickness	T_major_vein_	μm	***	***	ns
Adaxial stomatal pore length	PL_ad_	μm	***	***	***
Abaxial stomatal pore length	PL_ab_	μm	***	*	***
Adaxial stomatal pore width	PW_ad_	μm	***	ns	ns
Abaxial stomatal pore width	PW_ab_	μm	***	*	**
Adaxial stomatal complex width	SW_ad_	μm	***	ns	*
Abaxial stomatal complex width	SW_ab_	μm	***	ns	**
Adaxial guard cell width	GCW_ad_	μm	***	ns	ns
Abaxial guard cell width	GCW_ab_	μm	***	ns	ns
Leaf width	LW	μm	***	**	*
Stomatal ratio	SR	-	***	ns	ns
Major vein density	VLA_major_	mm mm^−2^	***	ns	ns
Minor vein density	VLA_minor_	mm mm^−2^	***	**	**
Inter-vein distance	IVD	μm	**	*	ns
Light saturated photosynthetic rate with adaxial illuminated	A_sat_ad_	μmol m^−2^ s^−1^	***	***	***
Light saturated photosynthetic rate with abaxial illuminated	A_sat_ab_	μmol m^−2^ s^−1^	***	**	***
Photosynthetic rate with adaxial illuminated	A_ad1800_	μmol m^−2^ s^−1^	***	***	*
Photosynthetic rate with abaxial illuminated	A_ab1800_	μmol m^−2^ s^−1^	***	***	*
Stomatal conductance with adaxial illuminated	g_sw_ad1800_	mol m^−2^ s^−1^	***	*	ns
Stomatal conductance with abaxial illuminated	g_sw_ab1800_	mol m^−2^ s^−1^	***	***	ns

## Data Availability

The datasets presented in the study are either included in the article or in the Appendix A; further inquiries can be directed to the corresponding authors.

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
