# Peer review of "Stomatal Ratio Showing No Response to Light Intensity in Oryza"

_plants, 2022, doi:10.3390/plants12010066_

Round 1
Reviewer 1 Report
This manuscript reports the robust stomatal ratios between 5 Oryza species grown in two different growth conditions. Basically, the manuscripts read well, and the authors analyzed data sufficiently especially leaf morphological traits. I still have some comments to improve the manuscript more impactful.
The major finding of this research is the robust stomatal ratio in Oryza species irrespective of growth environments mainly due to different light intensities. However, Authors mentioned the background of this point in a few sentences (e.g. Line 71). My major concern is that the authors failed to address why is the stomatal ratio robust in monocots and the what is the difference between the robust leaf traits (e.g. major vein density and guard cell width) and other leaf traits.
The title, “no response to growth environments” is a bit exaggerated because only two different growth conditions were used in this study, and the main difference in growth conditions is light intensity.
Line 10-11: The sentence is too vague. Please clarify what is unclear.
Line33: gs–>gs
Line38, 39: guard cells–>stomata (better to unify)
Line40: gs–>gs (many mistakes in subscript and superscript)
Line43: delete “F”
Line52, 54-55: Following two sentences are repetitive. “Stomatal size and density are sensitive〜” and “Stomatal size and density were also dramatically impacted〜”
Line186: Why the stomatal conductance is large enough to support photosynthesis? For example, in O. punctata, the smaller gs in outdoor growth condition reduced A as shown in fig. S4.
Line198: Awkward English sentence.
Line204: If you say “the stomatal density of Oryza species cannot be identified in the current study”, you should discuss other environmental factors other than light intensity with previous studies, at least air temperature.
Line 235-236: Add some references.
Author Response
RESPONSE TO REVIEWERS COMMENTS
Reviewer #1
We appreciate the positive comments from the reviewer about our general effort and the significance of this study for plasticity responses of stomatal traits to growth conditions, and his/her support for publication. We have carefully considered the suggestion of reviewer and make some changes.
This manuscript reports the robust stomatal ratios between 5 Oryza species grown in two different growth conditions. Basically, the manuscripts read well, and the authors analyzed data sufficiently especially leaf morphological traits. I still have some comments to improve the manuscript more impactful.
Q: The major finding of this research is the robust stomatal ratio in Oryza species irrespective of growth environments mainly due to different light intensities. However, Authors mentioned the background of this point in a few sentences (e.g. Line 71). My major concern is that the authors failed to address why is the stomatal ratio robust in monocots and the what is the difference between the robust leaf traits (e.g. major vein density and guard cell width) and other leaf traits.
A: The stomata on Oryza leaves distribute in parallel rows along the veins in both adaxial and abaxial surfaces(see the following slices).While there is always one row in adaxial surfaces and two rows in abaxial surfaces, the robust distribution pattern contributed to the robust stomatal ratio.
The second question the reviewer proposed was quite meaningful for the differences or the connection between the robust leaf traits and other leaf traits was almost no in-depth analysis in the manuscipts. So we discussed it here and added text elaborating on this of revised MS.
We divided these leaf traits into demand related traits and supply related according to their function in photosynthetic gas exchange. The demand traits like SDab and SDad were found higher in outdoor condition than growth chamber, which means that gas exchange demand was increased under high light. Considering neither major nor minor vein density contributed to the balance between supply and demand and inter-vein distance was increased under high light, we concluded that it was the increased vein thickness that worked. The decrease of the minor vein density, the increase of the leaf width and the inter-vein distance together indicated that the mesophyll tissue was increased by the wider distance between two veins and higher transportation capacity of each conduit. But the unincreased vein density in those five species indicate the existence of vascular redundancy, which means overweight in leaf vein. (see Line 235-254 of revised MS)
Besides, we have measured the major vein density of all the rice species used here (see the figure below) and found a stable relationship between leaf vein density and leaf width whether under field or green house condition, which is consistent with the results of our experiment.
Q: The title, “no response to growth environments” is a bit exaggerated because only two different growth conditions were used in this study, and the main difference in growth conditions is light intensity.
A: We quite agree with your opinion, and the title has been changed as “Stomatal ratio showing no response to light intensity in Oryza”.
Q: Line 10-11: The sentence is too vague. Please clarify what is unclear.
A: We have reorganized the sentence. (see Line 10-11 of revised MS)
Q: Line33: gs–>gs
A: Done. (see Line 33 of revised MS)
Q: Line38, 39: guard cells–>stomata (better to unify)
A: We have unified the expression here. (see Line 41 of revised MS)
Q: Line40: gs–>gs (many mistakes in subscript and superscript)
A: Following reviewer’s suggestion, the mistakes in the manuscripts were checked and revised.
Q: Line43: delete “F”
A: Done. (see Line 43 of revised MS)
Q: Line52, 54-55: Following two sentences are repetitive. “Stomatal size and density are sensitive〜” and “Stomatal size and density were also dramatically impacted〜”
A: Following reviewer’s suggestion, we elaborate more on the text to make this point explicit. (see Line 56-58 of revised MS)
Q: Line186: Why the stomatal conductance is large enough to support photosynthesis? For example, in O. punctata, the smaller gs in outdoor growth condition reduced A as shown in fig. S4.
A: The stomatal conductance was relatively high (>0.4 μ mol m-2 s-1) in all the estimated species in two growth conditions, which is considered high stomatal conductance for pot plants. That is why we said that “the stomatal conductance is large enough to support photosynthesis”. As for the fig. S4 you mentioned, the smaller gs for O. punctat was found to have different behaviors as others because this species reached high A even under lower gs when PAR is 1800 μ mol m-2 s-1, which also showed that the stomatal conductance is large enough to support photosynthesis.
Q: Line198: Awkward English sentence.
A: The following sentence is the new version:Salisbury[37]first reported that the stomatal density of the same species was higher in plants grown in sun conditions than in shade conditions in 1928. (see Line 209-210 of revised MS)
Q: Line204: If you say “the stomatal density of Oryza species cannot be identified in the current study”, you should discuss other environmental factors other than light intensity with previous studies, at least air temperature.
A: We reorganized the sentence here and discussed other environmental factors before trying to identify the main factors. ( see Line 216-217 of revised MS)
Q: Line 235-236: Add some references.
A: Done, the new added references are as following:
- Paradiso, R.; Marcelis, L.F.M. The Effect of Irradiating Adaxial or Abaxial Side on Photosynthesis of Rose Leaves. Acta Horticulturae 2012, 956, 157-163.
- Wang, C.; Du, Y.M.; Zhang, J.X.; Ren, J.T.; He, P.; Wei, T.; Xie, W.; Yang, H.K. Effects of exposure of the leaf abaxial surface to direct solar radiation on the leaf anatomical traits and photosynthesis of soybean (Glycine max L.) in dryland farming systems. Photosynthetica 2021, 59, 496-507.

Reviewer 2 Report
Line 258: Five Oryza species which have significant difference in stomatal traits…Who said that? Please, place a valid citation for this.
How many plants per species were used? 16?
Line 261: …one plant per pot … Which was the distance between the pots?
Line 270-271: …measurements began to conduct fifty days after planting. Why has been chosen this data, fifty days after planting? Please, explain better.
Line 296: …was conducted in Image J software (National Institute of Health). That is, where? Where is this Nation Institute of Health?
Line 103: Figure 1 Photosynthtic…Please, put the dot after Figure 1. And the same at Figure 2.
Author Response
RESPONSE TO REVIEWERS COMMENTS
Reviewer #2:
We appreciate the comments from the reviewer about our study for plasticity responses of stomatal traits to growth conditions, and his/her support for publication. We have carefully considered the suggestion of reviewer and make some changes.
Q: Line 258: Five Oryza species which have significant difference in stomatal traits…Who said that? Please, place a valid citation for this.
A: Before the experiment, we have measured the stomatal density of 47 Oryza species and chose the following five Oryza species which have significant difference in stomatal traits of both leaf sides. The figure attached are the stomatal traits of 47 Oryza species and the chosen Oryza species. O. australiensis, O. glumaepatala, O. minuta, O. punctata correspond to W4, W13, W22 and W34 in the first figure. Following reviewer’s suggestion, we added text elaborating on this of revised MS. (see Line 290 of revised MS)
Q: How many plants per species were used? 16?
A: Yes, 16 plants per species were used and half of them were placed outdoor and another half were placed in the growth chamber. The information had been added. (see Line 294 of revised MS)
Q: Line 261: …one plant per pot … Which was the distance between the pots?
A: The distance between the pots was 40cm, and we had added the content to the manuscript. (see Line 297 of revised MS)
Q: Line 270-271: …measurements began to conduct fifty days after planting. Why has been chosen this data, fifty days after planting? Please, explain better.
A: O. australiensis, O. glumaepatala, O. minuta, and O. punctata used here are wild rice, we had to take small tillers from the perennial plants to have new individuals. It taken time (about 50 days) for small tillers to take strong roots and grow to the active tillering stage. (see Line 309 of revised MS)
Q: Line 296: …was conducted in Image J software (National Institute of Health). That is, where? Where is this Nation Institute of Health?
A: Nation Institute of Health is the institution who first invented the software, but the citation we used here was wrong. The right citation has been changed as following: Image analysis was conducted in Image J software (National Institute of Health, Bethesda, MD, USA). (see Line 334 of revised MS)
Q: Line 103: Figure 1 Photosynthtic…Please, put the dot after Figure 1. And the same at Figure 2.
A: Thanks for pointing out this obvious error. Done.
